# Derivation and internal validation of prediction models for pulmonary hypertension risk assessment in a cohort inhabiting Tibet, China

Junhui Tang[1†], Rui Yang[2†], Hui Li[1], Xiaodong Wei[1], Zhen Yang[1], Wenbin Cai[1], Yao Jiang[1], Ga Zhuo[1], Li Meng[1], Yali Xu[3]*

[1]Department of Ultrasound, the General Hospital of Tibet Military Command, Tibet, China; [2]Department of High Mountain Sickness, the General Hospital of Tibet Military Command, Tibet, China; [3]Department of Ultrasound, Xinqiao Hospital, Army Medical University, Chongqing, China

## eLife Assessment

This study retrospectively analyzed clinical data to develop a risk prediction model for pulmonary hypertension in high-altitude populations. The evidence is **solid**, and the findings are **useful** and hold clinical significance as the model can be used for intuitive and individualized prediction of pulmonary hypertension risk in these populations.

**\*For correspondence:**
xuyali1976@163.com

[†]These authors contributed equally to this work

**Competing interest:** The authors declare that no competing interests exist.

**Abstract** Individuals residing in plateau regions are susceptible to pulmonary hypertension (PH) and there is an urgent need for a prediction nomogram to assess the risk of PH in this population. A total of 6603 subjects were randomly divided into a derivation set and a validation set at a ratio of 7:3. Optimal predictive features were identified through the least absolute shrinkage and selection operator regression technique, and nomograms were constructed using multivariate logistic regression. The performance of these nomograms was evaluated and validated using the area under the curve (AUC), calibration curves, the Hosmer–Lemeshow test, and decision curve analysis. Comparisons between nomograms were conducted using the net reclassification improvement (NRI) and integrated discrimination improvement (IDI) indices. Nomogram[I] was established based on independent risk factors, including gender, Tibetan ethnicity, age, incomplete right bundle branch block (IRBBB), atrial fibrillation (AF), sinus tachycardia (ST), and T wave changes (TC). The AUCs for Nomogram[I] were 0.716 in the derivation set and 0.718 in the validation set. Nomogram[II] was established based on independent risk factors, including Tibetan ethnicity, age, right axis deviation, high voltage in the right ventricle, IRBBB, AF, pulmonary P waves, ST, and TC. The AUCs for Nomogram[II] were 0.844 in the derivation set and 0.801 in the validation set. Both nomograms demonstrated satisfactory clinical consistency. The IDI and NRI indices confirmed that Nomogram[II] outperformed Nomogram[I]. Therefore, the online dynamic Nomogram[II] was established to predict the risks of PH in the plateau population.

## Introduction

Pulmonary hypertension (PH) is a chronic, progressive condition characterised by elevated pulmonary arterial pressure, primarily resulting from pulmonary vascular remodelling. This remodelling is driven by the infiltration of inflammatory cells, endothelial-to-mesenchymal transition, and hyperplasia of the

pulmonary intima (*Rubin and Naeije, 2023*; *Shah et al., 2023*; *Simonneau et al., 2019*). PH often presents similarly to other lung diseases, leading to diagnostic delays and, consequently, delays in receiving optimal treatment. Approximately 1% of the adult population and more than half of individuals with congestive heart failure are affected by PH (*Hoeper et al., 2016*; *Mandras et al., 2020*). Moreover, as the pulmonary vascular load increases, PH can ultimately lead to life-threatening right heart failure. The 1- and 3-year survival rates for patients with PH range from 68% to 93% and 39% to 77%, respectively (*Naeije et al., 2022*; *Ruopp and Cockrill, 2022*).

Right heart catheterisation (RHC) is recognised as the gold standard for diagnosing PH, clarifying the specific diagnosis, and determining the severity of the condition. However, due to its invasive nature, RHC is not suitable as a widespread population screening tool for PH (*McGoon et al., 2004*). Transthoracic echocardiography (TTE), a non-invasive screening test, is extensively used for PH because it can provide estimates of pulmonary arterial systolic pressure (sPAP) and evaluates cardiac structure and function. A clinical study involving 731 patients in China found no significant difference between RHC and TTE in assessing sPAP in PH caused by hypoxia. Furthermore, Pearson correlation analysis between RHC and TTE demonstrated a moderate overall correlation (*Hong et al., 2023*; *McGoon et al., 2004*; *Xu and Jing, 2009*).

According to literature reviews, nearly 140 million individuals reside in high-altitude regions (altitudes exceeding 2500 m), and the number of people visiting these areas for economic or recreational reasons has been increasing over the past few decades (*Moore et al., 1998*; *West, 2012*; *Xu and Jing, 2009*). High altitude typically signifies a hypoxic environment due to the decrease in barometric pressure as altitude increases, which proportionally reduces $PO_2$, resulting in hypobaric hypoxia (*Gassmann et al., 2021*). PH arising from prolonged exposure to hypoxic conditions at high altitudes is termed high-altitude PH (*Xu and Jing, 2009*). Hypoxia triggers hypoxic pulmonary vasoconstriction (HPV), a physiological response aimed at optimising ventilation/perfusion matching by redirecting blood to better-oxygenated segments of the lung through the constriction of small pulmonary arteries (*Dunham-Snary et al., 2017*). Furthermore, sustained hypoxia leads to pulmonary vascular remodelling, increasing resistance to blood flow due to reduced vessel elasticity and decreased vessel diameter. HPV and vascular remodelling are the primary mechanisms underlying hypoxia-induced PH, which significantly impairs right ventricular function and can ultimately result in fatal heart failure (*Julian and Moore, 2019*; *Penaloza and Arias-Stella, 2007*). Consequently, there is a pressing need for a straightforward and dependable model to assist clinicians and individuals in assessing the risk of PH in populations at high altitudes.

In this study, we developed and validated two risk prediction models for high-altitude PH based on TTE results by examining routine inspection parameters in Tibet, China.

## Results

### Subjects' characteristics

Following a 7:3 allocation ratio, 4622 subjects were placed in the derivation set and 1981 subjects in the validation set. The characteristics of the subjects are presented in *Table 1*. The prevalence of PH of Grade I or higher was 39.57% (1829 cases) in the derivation set and 39.27% (778 cases) in the validation set (p = 0.820 > 0.05). The prevalence of PH of Grade II or higher was 8.55% (395 cases) in the derivation set and 8.58% (170 cases) in the validation set (p = 0.962 > 0.05). No significant difference was observed in the age distribution between the derivation and validation sets (42.43 ± 16.93 vs 42.05 ± 16.41, p = 0.390 > 0.05), with age categorised into ≤42 and >42 subgroups based on the mean age. The composition ratios of the two age subgroups did not significantly differ between the validation and derivation sets (p = 0.6352 > 0.05). Furthermore, no significant differences were observed in the characteristics related to gender, Tibetan or not, right axis deviation (RAD), clockwise rotation (CR), counterclockwise rotation (CCR), high voltage in the right ventricle (HVRV), incomplete right bundle branch block (IRBBB), complete right bundle branch block (CRBBB), atrial fibrillation (AF), sinus arrhythmia (SA), sinus tachycardia (ST), sinus bradycardia (SB), T wave changes (TC), ST-segment changes (STC), atrial premature beats (APB), ventricular premature beats (VPB), junctional premature beats (JPB), pulmonary P waves (PP), atrioventricular block (IAB, I-degree atrioventricular block), and complete left bundle branch block (CLBBB) (*Table 1*).

**Table 1.** Baseline characteristics of individuals in the derivation and validation sets.

| Variable | Derivation set (*n* = 4622) | Validation set (*n* = 1981) | p |
|---|---|---|---|
| Age Total (mean ± SD) | 42.43 ± 16.93 | 42.05 ± 16.41 | 0.390 |
| Age ≤42, *n* (%) | 2619 (56.66) | 1135 (57.29) | |
| Age >42, *n* (%) | 2003 (43.34) | 846 (42.71) | 0.635 |
| | | | |
| Tibetan, *n* (%) | | | 0.538 |
| No | 2856 (61.79) | 1240 (62.59) | |
| Yes | 1766 (38.21) | 741 (37.41) | |
| Gender, *n* (%) | | | 0.260 |
| Female | 1219 (26.37) | 549 (27.71) | |
| Male | 3403 (73.63) | 1432 (72.29) | |
| RAD, *n* (%) | | | 0.141 |
| No | 3833 (82.93) | 1672 (84.40) | |
| Yes | 789 (17.07) | 309 (15.60) | |
| CR, *n* (%) | | | 0.387 |
| No | 4000 (86.54) | 1730 (87.33) | |
| Yes | 622 (13.46) | 251 (12.67) | |
| CCR, *n* (%) | | | 0.402 |
| No | 3994 (86.41) | 1727 (87.18) | |
| Yes | 628 (13.59) | 254 (12.82) | |
| HVRV, *n* (%) | | | 0.102 |
| No | 4151 (89.81) | 1805 (91.12) | |
| Yes | 471 (10.19) | 176 (8.88) | |
| IRBBB, *n* (%) | | | 0.573 |
| No | 4547 (98.38) | 1945 (98.18) | |
| Yes | 75 (1.62) | 36 (1.82) | |
| CRBBB, *n* (%) | | | 0.945 |
| No | 4444 (96.15) | 1904 (96.11) | |
| Yes | 178 (3.85) | 77 (3.89) | |
| AF, *n* (%) | | | 0.594 |
| No | 4551 (98.46) | 1954 (98.64) | |
| Yes | 71 (1.54) | 27 (1.36) | |
| SA, *n* (%) | | | 0.243 |
| No | 4247 (91.89) | 1837 (92.73) | |
| Yes | 375 (8.11) | 144 (7.27) | |
| ST, *n* (%) | | | 0.910 |
| No | 4395 (95.09) | 1885 (95.15) | |
| Yes | 227 (4.91) | 96 (4.85) | |
| SB, *n* (%) | | | 0.345 |
| No | 4245 (91.84) | 1833 (92.53) | |
| Yes | 377 (8.16) | 148 (7.47) | |
| TC, *n* (%) | | | 0.769 |
| No | 4003 (86.61) | 1721 (86.88) | |

*Table 1 continued on next page*

*Table 1 continued*

| Variable | Derivation set (*n* = 4622) | Validation set (*n* = 1981) | p |
|---|---|---|---|
| Yes | 619 (13.39) | 260 (13.12) | |
| STC, *n* (%) | | | 0.415 |
| No | 4399 (95.18) | 1876 (94.70) | |
| Yes | 223 (4.82) | 105 (5.30) | |
| APB, *n* (%) | | | 0.219 |
| No | 4587 (99.24) | 1960 (98.94) | |
| Yes | 35 (0.76) | 21 (1.06) | |
| JPB, *n* (%) | | | 0.425 |
| No | 4603 (99.59) | 1970 (99.44) | |
| Yes | 19 (0.41) | 11 (0.56) | |
| VPB, *n* (%) | | | 0.844 |
| No | 4580 (99.09) | 1962 (99.04) | |
| Yes | 42 (0.91) | 19 (0.96) | |
| PP, *n* (%) | | | 0.439 |
| No | 4507 (97.51) | 1938 (97.83) | |
| Yes | 115 (2.49) | 43 (2.17) | |
| CLBBB, *n* (%) | | | 0.757 |
| No | 4610 (99.74) | 1975 (99.70) | |
| Yes | 12 (0.26) | 6 (0.30) | |
| IAB, *n* (%) | | | 0.910 |
| No | 4556 (98.57) | 1952 (98.54) | |
| Yes | 66 (1.43) | 29 (1.46) | |
| PH ≥ I grade, *n* (%) | | | 0.820 |
| No | 2793 (60.43) | 1203 (60.73) | |
| Yes | 1829 (39.57) | 778 (39.27) | |
| PH ≥ II grade, *n* (%) | | | 0.962 |
| No | 4227 (91.45) | 1811 (91.42) | |
| Yes | 395 (8.55) | 170 (8.58) | |

The online version of this article includes the following source data for table 1:

**Source data 1.** The raw data of *Table 1*.

## Independent risk factors in PH ≥ I grade group and PH ≥ II grade group

In the PH ≥ I grade group, based on the $\lambda\_min$ criterion in the least absolute shrinkage and selection operator (LASSO) regression model, 18 out of 22 variables were selected. However, this selection was deemed excessive for practical clinical applications. Therefore, we further refined the model using the $\lambda\_1se$ criterion, which reduced the number of variables, albeit with a significant decrease in the area under the curve (AUC) of the receiver operating characteristic (ROC) curve ($\lambda\_1se$) compared to the ROC curve ($\lambda\_min$) (*Figures 1 and 2C, E, and G*). Ultimately, nine variables were chosen according to $\lambda\_1se$, including gender, Tibetan ethnicity, age ≤42, age >42, IRBBB, CRBBB, AF, ST, and TC (*Figure 2I*). Gender, Tibetan ethnicity, age, IRBBB, AF, ST, and TC were subsequently identified as independent risk factors for PH ≥ I grade through multivariate logistic regression analysis and were used to develop Nomogram[I] (*Table 2*).

In the PH ≥ II grade group, based on the $\lambda\_1se$ criterion in the LASSO regression model (*Figure 2D, F*), 11 variables were selected to align with clinical needs. These variables were Tibetan ethnicity, age ≤42, age >42, RAD, HVRV, IRBBB, CRBBB, AF, PP, ST, and TC (*Figure 2J*). Tibetan ethnicity, age, RAD, HVRV, IRBBB, AF, PP, ST, and TC were determined to be independent risk factors for PH ≥ II

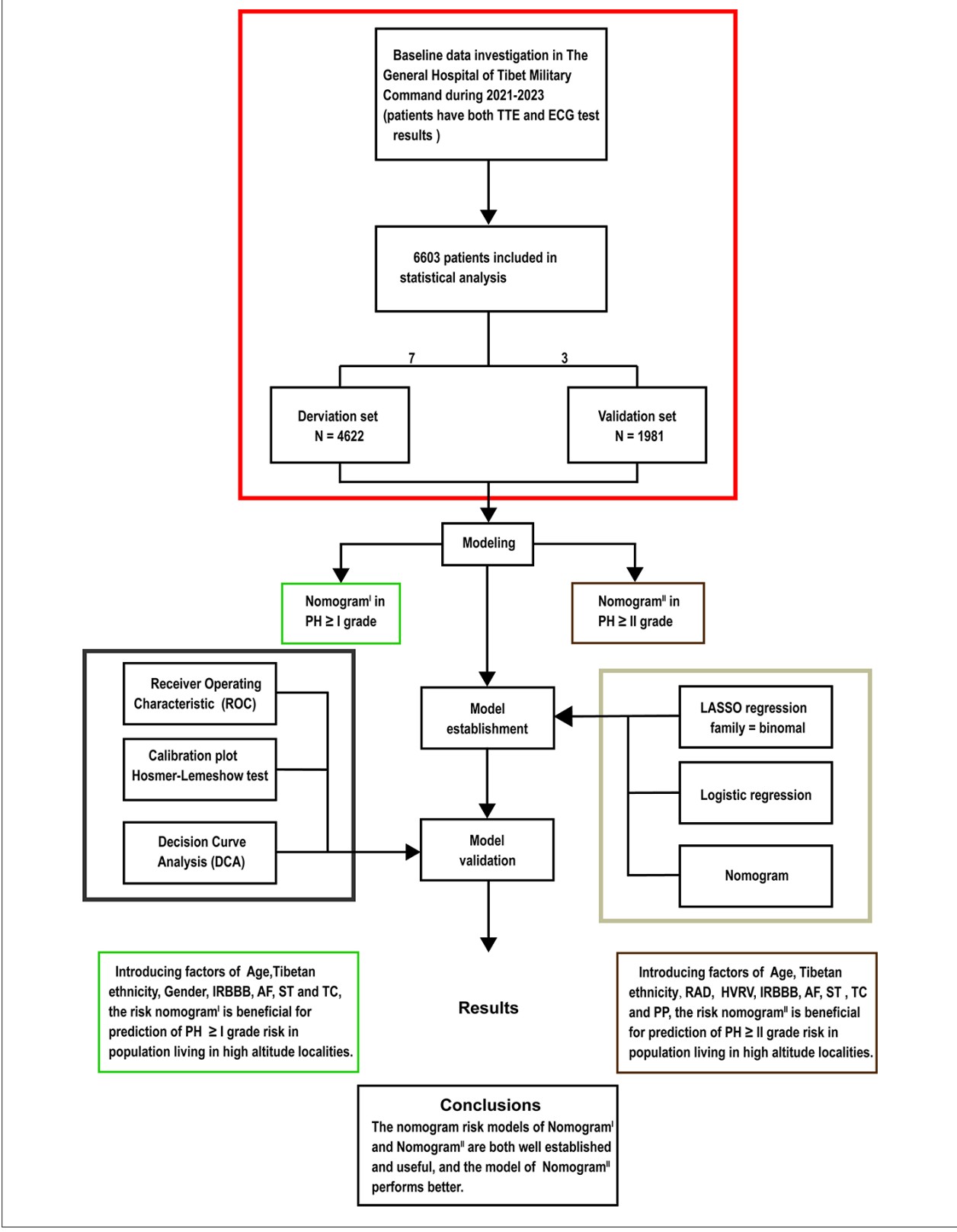

**Figure 1.** Flow diagram. Based on the exclusion and inclusion criteria, 6603 patients were included in this study. Patients were divided into a validation set and a derivation set randomly following a 7:3 ratio. Pulmonary hypertension, PH; right axis deviation, RAD; high voltage in the right ventricle, HVRV; incomplete right bundle branch block, IRBBB; atrial fibrillation, AF; sinus tachycardia, ST; T wave changes, TC; pulmonary P waves, PP.

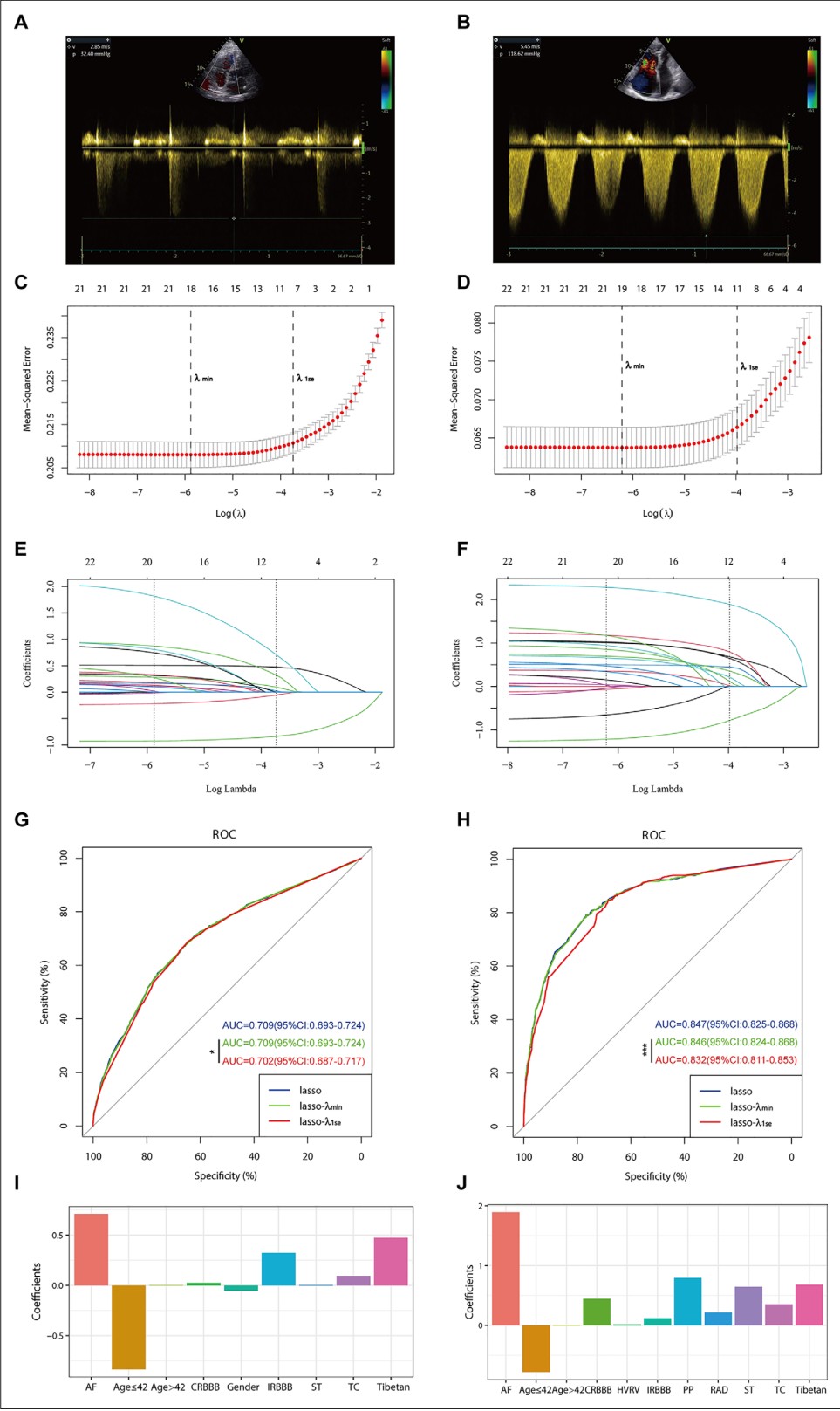

**Figure 2.** Illustrates the optimal predictive variables as determined by the least absolute shrinkage and selection operator (LASSO) binary logistic regression model. Panels A and B depict the measurement of tricuspid regurgitation spectra via transthoracic echocardiography in patients with Grade I pulmonary hypertension (PH) (**A**) and Grade III PH (**B**). Panels C to J demonstrate the identification of the optimal penalisation coefficient lambda

*Figure 2 continued on next page*

*Figure 2 continued*

(λ) in the LASSO model using 10-fold cross-validation for the PH ≥ I grade group (**C**) and the PH ≥ II grade group (**D**). The dotted line on the left (λ_min) represents the value of the harmonic parameter log(λ) at which the model's error is minimised, and the dotted line on the right (λ_1se) indicates the value of the harmonic parameter log(λ) at which the model's error is minimal minus 1 standard deviation. The LASSO coefficient profiles of 22 predictive factors for the PH ≥ I grade group (**E**) and the PH ≥ II grade group (**F**) show that as the value of λ decreased, the degree of model compression increased, enhancing the model's ability to select significant variables. Receiver operating characteristic (ROC) curves were constructed for three models (LASSO, LASSO-λ_min, and LASSO-λ_1se) in both the PH ≥ I grade group (**G**) and the PH ≥ II grade group (**H**). Histograms depict the final variables selected according to λ_1se and their coefficients for the PH ≥ I grade group (**I**) and the PH ≥ II grade group (**J**). Asterisks denote levels of statistical significance: $*p < 0.05$, $**p < 0.01$, $***p < 0.001$.

The online version of this article includes the following source data for figure 2:

**Source data 1.** Raw data of *Figure 2*.

grade through multivariate logistic regression analysis and were utilised to construct Nomogram[II] (*Table 3*).

## Construction of Nomogram[I] in PH ≥ I grade group and Nomogram[II] in PH ≥ II grade group

In the PH ≥ I grade group, a predictive Nomogram[I] for PH ≥ I grade was developed based on independent risk factors, including gender, Tibetan ethnicity, age, IRBBB, AF, ST, and TC. Points are assigned to each independent factor by drawing a vertical line to the points scale. The total points for an individual correspond to their risk of developing PH. Patients were then classified into high- and low-risk subgroups according to the total score's cut-off value (cut-off value: 45), which was determined based on the ROC curve (*Figure 3A*). The risks for the two groups were evaluated in both the derivation and validation sets. In the derivation set, the risk of PH in the high-risk group was significantly higher than in the low-risk group (odds ratio [OR]: 4.210, 95% confidence interval [CI]: 3.715–4.775) (*Figure 3B*), as was also observed in the validation set (OR: 4.207, 95% CI: 3.476–5.102) (*Figure 3C*).

In the PH ≥ II grade group, a predictive Nomogram[II] for PH ≥ II grade was developed using independent risk factors, including Tibetan ethnicity, age, RAD, HVRV, IRBBB, AF, PP, ST, and TC. Based on the cut-off value of the total score (cut-off value: 76), determined in line with the ROC curve, patients were categorised into high- and low-risk subgroups (*Figure 3D*). The risks for the two groups were evaluated in both the derivation and validation sets. In the derivation set, the risk of PH in the high-risk group was significantly greater than in the low-risk group (OR: 11.591, 95% CI: 9.128–14.845) (*Figure 3E*), a finding that was replicated in the validation set (OR: 7.103, 95% CI: 5.106–9.966) (*Figure 3F*).

## Assessment and validation of Nomogram[I] in the PH ≥ I grade group and Nomogram[II] in the PH ≥ II grade group

In the PH ≥ I grade group, Nomogram[I] was developed to predict the risk of PH ≥ I grade, utilising the AUC to assess its discriminative ability. The AUC value for Nomogram[I] was 0.716 (95% CI: 0.701–0.731)

**Table 2.** Risk factors for pulmonary hypertension (PH) ≥ I grade in the derivation set.

| Variable | β-Coefficient | OR (95% CI) | p |
|---|---|---|---|
| Tibetan | 0.34 | 1.40 (1.23–1.60) | <0.001 |
| Gender | −0.3 | 0.74 (0.65–0.84) | <0.001 |
| Age | 0.034 | 1.03 (1.03–1.04) | <0.001 |
| IRBBB | 1.106 | 3.02 (1.96–4.67) | <0.001 |
| AF | 1.431 | 4.18 (2.19–7.97) | <0.001 |
| ST | 0.369 | 1.45 (1.14–1.84) | 0.003 |
| TC | 0.306 | 1.36 (1.16–1.59) | <0.001 |

The online version of this article includes the following source data for table 2:

**Source data 1.** Raw data of *Table 2*.

**Table 3.** Risk factors for pulmonary hypertension (PH) ≥ II grade in the derivation set.

| Variable | β-Coefficient | OR (95% CI) | P |
|---|---|---|---|
| Tibetan | 0.689 | 1.99 (1.55–2.57) | <0.001 |
| Age | 0.042 | 1.04 (1.03–1.05) | <0.001 |
| RAD | 0.751 | 2.12 (1.56–2.88) | <0.001 |
| HVRV | 0.486 | 1.63 (1.14–2.31) | 0.007 |
| IRBBB | 1.512 | 4.53 (2.77–7.42) | <0.001 |
| AF | 2.102 | 8.18 (5.13–13.05) | <0.001 |
| ST | 1.247 | 3.48 (2.58–4.70) | <0.001 |
| TC | 0.592 | 1.81 (1.44–2.27) | <0.001 |
| PP | 1.486 | 4.42 (2.96–6.61) | <0.001 |

The online version of this article includes the following source data for table 3:

**Source data 1.** The raw data of *Table 3*.

in the derivation set (*Figure 4A*) and 0.718 (95% CI: 0.695–0.741) in the validation set (*Figure 4D*). Furthermore, ROC curves were used to compare the discriminative capacity of Nomogram[I] and single independent factors in predicting PH ≥ I grade. Notably, the AUC of Nomogram[I] was significantly higher than that of any single independent factor in the derivation (*Figure 4B, C*) and the validation set (*Figure 4E, F*). The calibration curves for the derivation set (*Figure 5A*) and the validation set (*Figure 5B*) demonstrated high agreement between predicted and actual values, indicating that Nomogram[I] accurately predicts PH ≥ I grade. The results of the Hosmer–Lemeshow test in both the derivation set (p = 0.109 > 0.05) and the validation set (p = 0.317 > 0.05) further confirmed the effective performance of Nomogram[I] (*Figure 5E*).

Nomogram[II] was developed to predict the risk of PH ≥ II grade. The AUC for Nomogram[II] was 0.844 (95% CI: 0.823–0.865) in the derivation set (*Figure 4G*) and 0.801 (95% CI: 0.763–0.838) in the validation set (*Figure 4J*). Furthermore, ROC curves were used to compare the discriminative capacity of Nomogram[II] and individual independent factors in predicting PH ≥ II grade. The AUC of Nomogram[II] was significantly higher than that of any single independent factor in the derivation set (*Figure 4H, I*) and the validation set (*Figure 4K, L*). The calibration curves for the derivation set (*Figure 5C*) and the validation set (*Figure 5D*) demonstrated high agreement between the predicted and actual values, indicating that Nomogram[II] accurately predicts PH ≥ II grade. Additionally, the results of the Hosmer–Lemeshow test in the derivation set (p = 0.377 > 0.05) and the validation set (p = 0.127 > 0.05) further confirmed the good performance of Nomogram[II] (*Figure 5F*).

## Clinical utility of Nomogram[I] and Nomogram[II]

In the PH ≥ I grade group, the clinical utility of Nomogram[I] for predicting the risk of PH ≥ I grade was assessed using decision curve analysis (DCA). This analysis revealed a significant net benefit with a threshold probability range of 20–91% in the derivation set (*Figure 6A*) and 14–74% in the validation set (*Figure 6B*). Moreover, the DCA curve from the derivation set indicated that the clinical predictive capability of Nomogram[I] surpassed that of any single independent factor, a finding that was corroborated in the validation set (*Figure 6C, D*).

In the PH ≥ II grade group, the clinical utility of Nomogram[II] for predicting the risk of PH ≥ II grade was evaluated using DCA, which showed a clear net benefit within the threshold probability range of 1–70% in the derivation set (*Figure 6E*) and 1–82% in the validation set (*Figure 6F*). Additionally, the DCA curve for the derivation set demonstrated that the clinical predictive effectiveness of Nomogram[II] exceeded that of any single independent factor, a conclusion that was also confirmed in the validation set (*Figure 6G, H*).

## Comparison between Nomogram[I] and Nomogram[II]

In the PH ≥ I grade group, when comparing Nomogram[I] to Nomogram[II], Nomogram[I] exhibited an integrated discrimination improvement (IDI) of −0.0012 (95% CI: −0.0032 to 0.0009, p = 0.2777), a categorical net reclassification improvement (NRI) of 0.0117 (95% CI: −0.0004 to 0.0237, p = 0.0575),

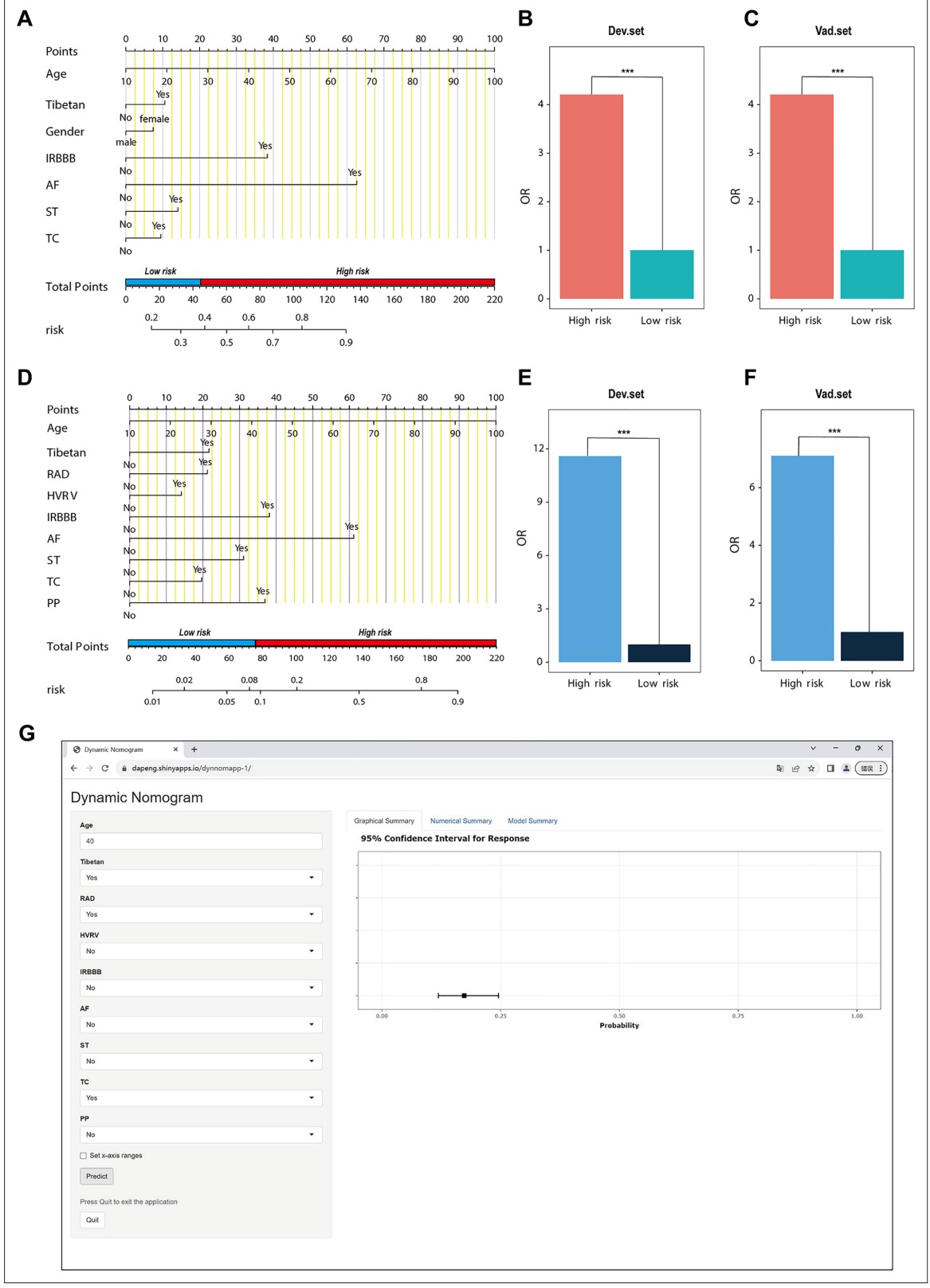

**Figure 3.** Nomogram for predicting pulmonary hypertension (PH) and risk stratification based on total score. (**A–C**) Nomogram[I] for the prediction of PH ≥ I grade in the PH ≥ I grade group. Points for each independent factor are summed to calculate total points, determining the corresponding 'risk' level. Patients were divided into 'High-risk' and 'Low-risk' subgroups according to the cut-off of the total points (**A**). Histograms illustrate the odds ratio (OR) comparing the 'High-risk' group to the 'Low-risk' group in the derivation set (**B**) and validation set (**C**). (**D–F**) Nomogram[II] for predicting PH ≥

*Figure 3 continued on next page*

*Figure 3 continued*

II grade within the PH ≥ II grade group: Similarly, points from each independent factor are totalled, and the corresponding 'risk' level is ascertained. Patients are divided into 'High-risk' and 'Low-risk' groups based on the cut-off value of the total points (**D**). Histograms display the OR for the 'High-risk' group compared to the 'Low-risk' group in the derivation (**E**) and validation set (**F**). \*\*\*p < 0.001. (**G**) Screenshot of dynamic Nomogram[II]'s web page.

The online version of this article includes the following source data for figure 3:

**Source data 1.** The raw data and R software code of *Figure 3*.

and a continuous NRI of −0.2423 (95% CI: −0.2992 to −0.1854, p < 0.001) in predicting the risk of PH ≥ I grade.

In the PH ≥ II grade group, compared to Nomogram[I], Nomogram[II] demonstrated an IDI of 0.0366 (95% CI: 0.0247–0.0485, p < 0.001), a categorical NRI of 0.0301 (95% CI: 0.0093–0.0510, p < 0.05), and a continuous NRI of 0.2785 (95% CI: 0.1824–0.3745, p < 0.001) for predicting the risk of PH ≥ II grade.

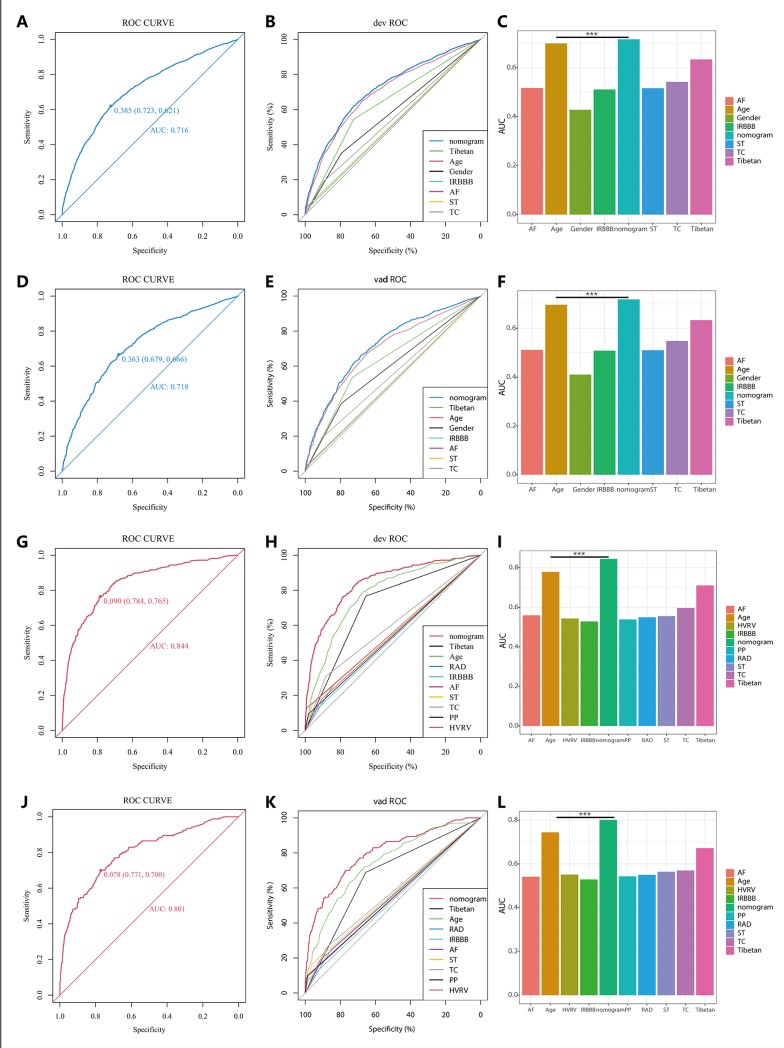

**Figure 4.** Receiver operating characteristic (ROC) curves and area under the curve (AUC) for Nomogram[I] in pulmonary hypertension (PH) ≥ I and Nomogram[II] in PH ≥ II grade groups. In the PH ≥ I grade group, the ROC and corresponding AUC of Nomogram[I] and independent factors in the derivation set (**A–C**) and validation set (**D–F**). In the PH ≥ II grade group, the ROC and corresponding AUC of Nomogram[II] and independent factors in the derivation set (**G–I**) and validation set (**J–L**).

The online version of this article includes the following source data for figure 4:

**Source data 1.** The raw data and R software code of *Figure 4*.

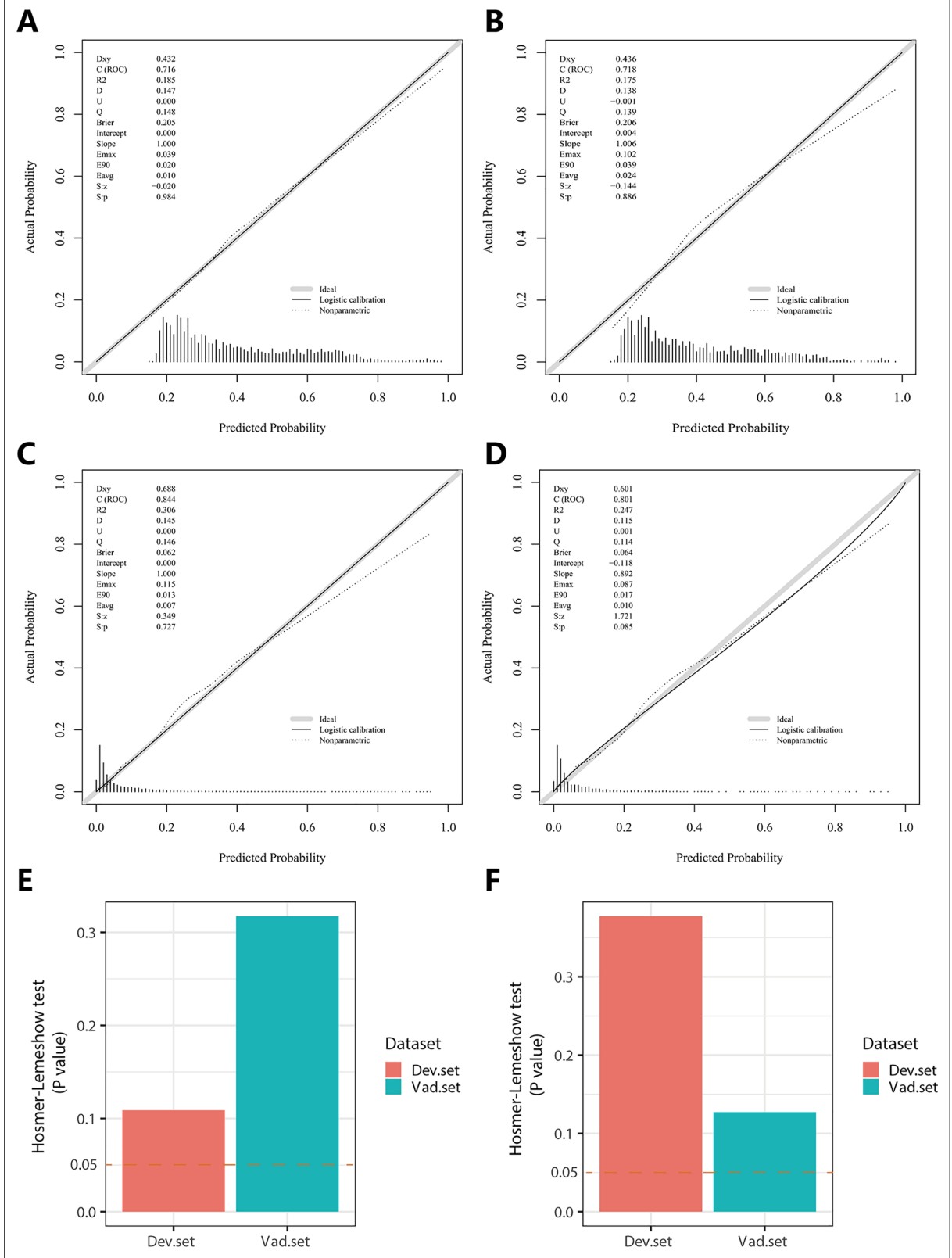

**Figure 5.** Calibration plots and Hosmer–Lemeshow test results for Nomogram$^I$ in pulmonary hypertension (PH) ≥ I and Nomogram$^{II}$ in PH ≥ II grade groups. In the PH ≥ I grade group, the calibration plots of Nomogram$^I$ in the derivation set (**A**) and the validation set (**B**). In the PH ≥ II grade group, the calibration plots of Nomogram$^{II}$ in the derivation set (**C**) and the validation set (**D**). (**E**) In the PH ≥ I grade group, Hosmer–Lemeshow test results for

*Figure 5 continued on next page*

*Figure 5 continued*

Nomogram[I] in the derivation set and the validation set. (**F**) In the PH ≥ II grade group, Hosmer–Lemeshow test results for Nomogram[II] in the derivation set and the validation set.

The online version of this article includes the following source data for figure 5:

**Source data 1.** The raw data and R software code of *Figure 5*.

These results indicate that Nomogram[II] outperformed Nomogram[I] in terms of IDI and NRI values.

## Website of Nomogram[II]

Patients and physicians can calculate the risk of PH through a free web-based dynamic Nomogram[II] (https://dapeng.shinyapps.io/dynnomapp-1/), and the screenshot of dynamic Nomogram[II]'s web page was shown (*Figure 3G*).

## Discussion

A significant portion of the global population lives in high-altitude areas such as the Tibetan Plateau, Ethiopian Highlands, Andes Mountains, and Pamir Plateau. These regions are marked by an extremely hypoxic environment that leads to alveolar hypoxia, posing severe risks to the cardiopulmonary system. One such risk is the development of PH, which occurs through mechanisms like HPV and pulmonary vascular remodelling (*Burtscher et al., 2018*; *Sydykov et al., 2021*; *Wilkins et al., 2015*). Accurate, timely diagnosis and early, effective treatment are crucial for the clinical improvement and survival of patients with PH. Without prompt intervention, PH can impair right heart function and ultimately result in fatal right heart failure (*Benza et al., 2010*; *Kim and George, 2019*; *McGoon et al., 2004*). Thus, there is a need to develop a predictive model to estimate the risk of PH, facilitating risk stratification and management. In this study, we analysed routine electrocardiogram (ECG) examination indicators and basic demographic information to assess the risk of PH. We developed nomograms for the PH ≥ I grade group and the PH ≥ II grade group, and the performance of these nomograms was evaluated and validated.

Currently, TTE is widely utilised for large-scale, non-invasive screening of patients at risk for PH (*D'Alto et al., 2018*; *Habib and Torbicki, 2010*; *Janda et al., 2011*). However, in plateau regions such as Tibet, medical resources are relatively limited, and remote villages and towns lack the facilities for TTE examinations. ECG examinations, being easy to administer, cost-effective, and feasible for remote delivery through telemedicine, offer a practical alternative (*Ismail et al., 2023*). A retrospective analysis has demonstrated that ECG examination results correlate with clinical parameters reflecting the severity of PH (*Michalski et al., 2022*). Therefore, in developing this model, we primarily relied on ECG examination results from patients. Utilising ECG results as predictors of PH can significantly aid clinicians in identifying potential PH patients in remote plateau areas, facilitating their access to timely and relevant treatment.

In this study, based on sPAP assessed by TTE examination, patients at risk of PH were classified into Grades I–III. We developed and validated two nomograms for the PH ≥ I grade group (Nomogram[I]) and the PH ≥ II grade group (Nomogram[II]), with ECG examination results serving as the primary component for both. Nomogram[I] included seven variables: gender, Tibetan ethnicity, age, IRBBB, AF, ST, and TC. Nomogram[II] incorporated nine variables: Tibetan ethnicity, age, RAD, HVRV, IRBBB, AF, PP, ST, and TC (*Figure 3A, D*). These variables are readily available from routine ECG examinations. Additionally, patients were categorised into high- and low-risk groups based on the cut-off value of the total score in the nomogram, with the OR value for the high-risk group being significantly higher than that of the low-risk group (*Figure 3*). Therefore, both nomograms offer a useful and straightforward method for in-depth evaluation, even without medical professional intervention. Both Nomogram[I] and Nomogram[II] demonstrated good calibration and clinical utility (*Figures 5 and 6*), though ROC analysis revealed that the AUC for Nomogram[II] was higher than that for Nomogram[I] (0.844 vs 0.716). IDI and NRI are recognised indicators that describe improved accuracy in predicting binary, multi-classification, or survival outcomes (*Wang et al., 2020*). In a similar vein to a 10-year retrospective cohort study, which constructed two nomograms for hypertension risk prediction and compared them using IDI and NRI values (*Deng et al., 2021*), we used IDI and NRI to evaluate the performance of Nomogram[I] and Nomogram[II]. Our findings indicated no significant difference between Nomogram[I]

and Nomogram[II] in the PH ≥ I grade group; however, Nomogram[II] exhibited superior performance compared to Nomogram[I] in the PH ≥ II grade group, thus demonstrating its enhanced predictive capability. So, we created an online dynamic Nomogram[II] for doctors and patients to calculate the risk of PH (*Figure 3G*).

In this study, age and Tibetan ethnicity were identified as independent predictors of PH, a finding that aligns with conclusions from a single-centre, cross-sectional study among native Tibetans in Sichuan Province, China (*Gou et al., 2020*). We hypothesise that this association may be due to the longer exposure to the hypoxic environment at high altitudes experienced by older individuals and Tibetans, promoting hypoxic contraction of pulmonary blood vessels and subsequent pulmonary vascular remodelling, thereby leading to PH. Additionally, the occurrence of AF emerged as an independent predictor of PH with the highest OR values in both nomograms (*Tables 2 and 3*). PH is known to be characterised by pulmonary vascular remodelling, which can induce fibrosis and excessive myocardial apoptosis, ultimately contributing to AF (*Yi et al., 2023*), a finding that corroborates our results. Nonetheless, it was observed that no single predictor alone was effective in distinguishing PH, exhibiting poor clinical utility compared to the comprehensive approach offered by the nomogram (*Figures 4 and 6*).

Our study has several limitations. Firstly, TTE serves only as a screening method for PH and is not the gold standard; its results merely indicate the risk of PH in the examined individuals. Secondly, given the constrained medical resources in remote areas, we primarily incorporated readily ECG results and basic demographic information into the nomograms, resulting in a relatively simple set of independent predictors. Lastly, the dataset for this study was exclusively sourced from Tibet, China, meaning the validation of the nomograms lacks external validation sets.

## Conclusion

We have developed a reliable and straightforward nomogram to predict the risks associated with PH, demonstrating satisfactory discrimination and calibration. Upon rigorous validation using internal datasets, the nomogram has shown clinical utility and favourable predictive accuracy. It is anticipated to serve as an effective and convenient clinical tool for assessing the risk of PH in populations residing at high altitudes.

## Materials and methods

### Study population and data collection

Upon gathering data from all patients who underwent both TTE and 12-lead ECG examinations at the General Hospital of Tibet Military Command between April 2021 and October 2023, we further screened the records based on the following criteria: (1) age >14 years; (2) interval between the TTE and ECG examinations <2 months, and (3) for patients with multiple TTE and/or ECG records, only the examination with the shortest interval between TTE and ECG was selected. Ultimately, we compiled examination data for 6603 eligible patients.

The retrospectively collected clinical data were categorised into two main groups: (1) demographic characteristics, including name, age, gender, and Tibetan ethnicity; (2) ECG results, encompassing RAD, CR, CCR, HVRV, IRBBB, CRBBB, AF, SA, SB, ST, TC, STC, APB, VPB, JPB, CLBBB, first-degree IAB, and PP; (3) TTE results: sPAP was measured via TTE to evaluate PH. PH was graded as follows: Grade I PH (50 mmHg > sPAP ≥ 30 mmHg), Grade II PH (70 mmHg > sPAP ≥ 50 mmHg), and Grade III PH (sPAP ≥ 70 mmHg). The severity of PH increases with its grade, indicating a higher risk of the condition.

All procedures were conducted following the approval of the Ethics Committee of the General Hospital of Tibet Military Command (Approval Number: 2024-KD002-01). Subsequently, the data from all participants were anonymised and de-identified prior to analysis. Consequently, the requirement for informed consent was waived.

### Statistical analysis

Statistical analysis was performed with R software version 4.3.2. $p < 0.05$ (double-tailed) was considered statistically significant.

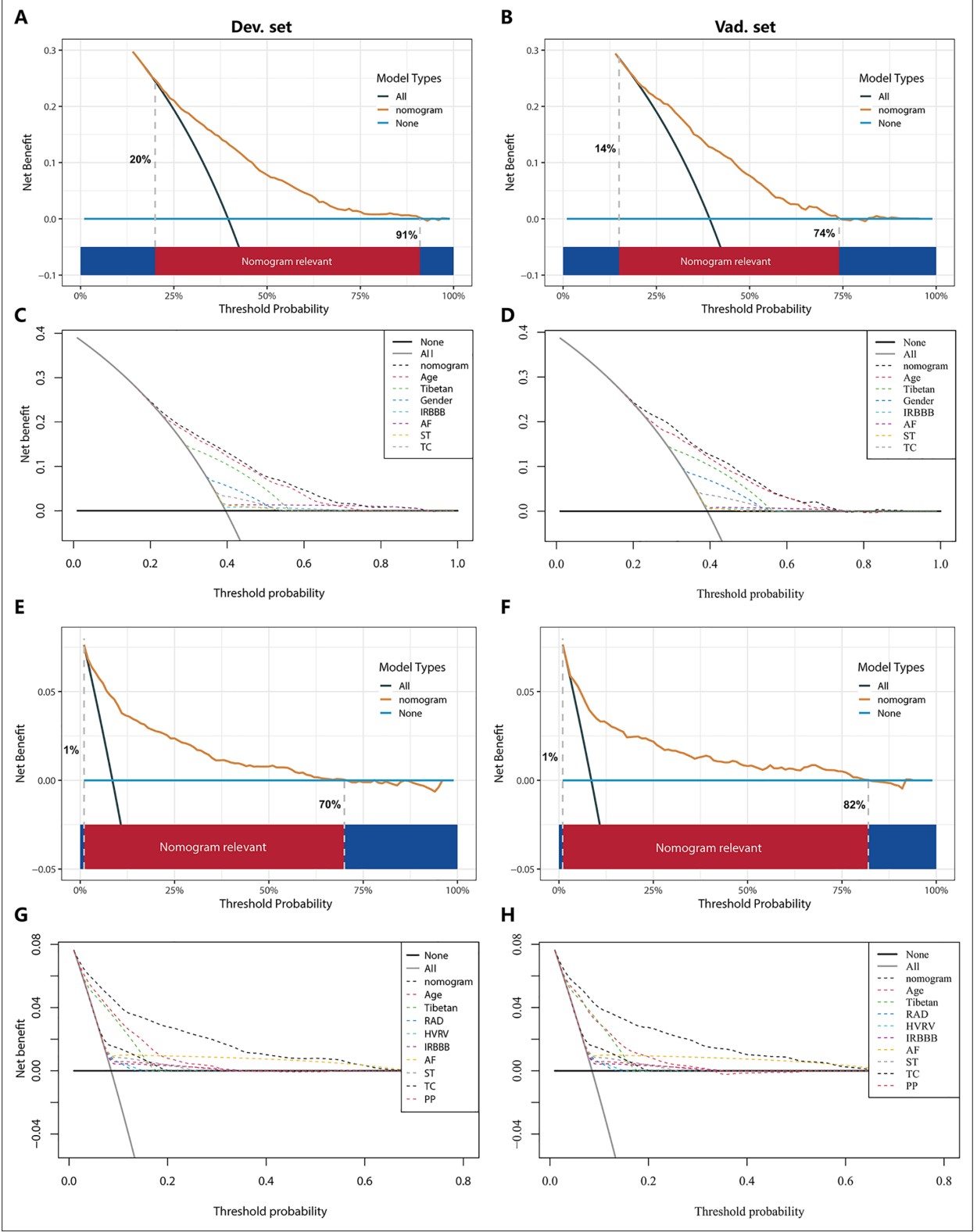

**Figure 6.** Decision curve analysis (DCA) for Nomogram[I] in the pulmonary hypertension (PH) ≥ I grade and Nomogram[II] in the PH ≥ II grade group. In the PH ≥ I grade group, the DCAs of Nomogram[I] and independent factors in the derivation (**A, C**) and validation set (**B, D**). In the PH ≥ II grade group, the DCAs of Nomogram[II] and independent factors in the derivation (**E, G**) and validation set (**F, H**).

The online version of this article includes the following source data for figure 6:

**Source data 1.** The raw data and R software code of *Figure 6*.

For validation and derivation of the prediction model, subjects were divided into a validation set and a derivation set randomly, at a ratio of 7:3, respectively. Categorical variables were transformed into dichotomous variables, and continuous variables were expressed by concrete values (means ± standard deviation) and analysed using Student's $t$-test. Fisher's exact test or Pearson's $\chi^2$ test was applied for categorical variables.

The derivation set was used to select optimal predictive factors through the LASSO regression technique. Independent factors were identified via multivariate logistic regression analysis, incorporating variables selected during the LASSO regression. A backward step-down selection process, guided by the Akaike information criterion, determined the final model. The predictive accuracy of the nomograms was assessed using the AUC of the ROC curve in both the derivation and validation sets. The Hosmer–Lemeshow test and calibration curves were employed to evaluate the consistency between actual outcomes and predicted probabilities. The clinical utility of the nomograms was assessed through DCA. The cut-off value for the total score in the nomogram was established based on the ROC curve, with patients categorised into low- and high-risk groups. The performance comparison between nomograms was analysed using the IDI and NRI.

## Acknowledgements

This study was funded by the Talent Program of Army Medical University (No. 2019R038).

## Additional information

### Funding

| Funder | Grant reference number | Author |
| --- | --- | --- |
| The Talent Program of Army Medical University | 2019R038 | Yali Xu |

The funders had no role in study design, data collection, and interpretation, or the decision to submit the work for publication.

### Author contributions

Junhui Tang, Conceptualization, Data curation, Software, Formal analysis, Methodology, Writing - original draft, Writing - review and editing; Rui Yang, Data curation, Software, Formal analysis, Project administration, Writing - review and editing; Hui Li, Conceptualization, Supervision; Xiaodong Wei, Data curation, Formal analysis; Zhen Yang, Wenbin Cai, Ga Zhuo, Data curation; Yao Jiang, Li Meng, Conceptualization; Yali Xu, Resources, Software, Funding acquisition, Validation, Investigation, Methodology, Writing - review and editing

### Author ORCIDs

Junhui Tang ⓘ http://orcid.org/0000-0001-6215-2913
Rui Yang ⓘ https://orcid.org/0000-0002-6949-7428
Yali Xu ⓘ https://orcid.org/0000-0002-6311-8757

### Ethics

All procedures were conducted following the approval of the Ethics Committee of the General Hospital of Tibet Military Area Command (Approval Number: 2024-KD002-01). Subsequently, the data from all participants were anonymised and de-identified prior to analysis. Consequently, the requirement for informed consent was waived.

Reviewer #1 (Public Review): https://doi.org/10.7554/eLife.98169.3.sa1
Author response https://doi.org/10.7554/eLife.98169.3.sa2

## Additional files

### Supplementary files
• MDAR checklist

### Data availability
All data generated or analysed during this study are included in the manuscript and supporting files.

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
