## [Editor Report · eLife Assessment]

This study retrospectively analyzed clinical data to develop a risk prediction model for pulmonary hypertension in high-altitude populations. The evidence is **solid**, and the findings are **useful** and hold clinical significance as the model can be used for intuitive and individualized prediction of pulmonary hypertension risk in these populations.

---

## [Referee Report · Reviewer #1 (Public Review)]

Summary:

This study retrospectively analyzed clinical data to develop a risk prediction model for pulmonary hypertension in high-altitude populations. This finding holds clinical significance as it can be used for intuitive and individualized prediction of pulmonary hypertension risk in these populations. The strength of evidence is high, utilizing a large cohort of 6,603 patients and employing statistical methods such as LASSO regression. The model demonstrates satisfactory performance metrics, including AUC values and calibration curves, enhancing its clinical applicability.

Strengths:

(1) Large Sample Size: The study utilizes a substantial cohort of 6,603 subjects, enhancing the reliability and generalizability of the findings.

(2) Robust Methodology: The use of advanced statistical techniques, including least absolute shrinkage and selection operator (LASSO) regression and multivariate logistic regression, ensures the selection of optimal predictive features.

(3) Clinical Utility: The developed nomograms are user-friendly and can be easily implemented in clinical settings, particularly in resource-limited high-altitude regions.

(4) Performance Metrics: The models demonstrate satisfactory performance, with strong AUC values and well-calibrated curves, indicating accurate predictions.

Weaknesses:

(1) Lack of External Validation: The models were validated internally, but external validation with cohorts from other high-altitude regions is necessary to confirm their generalizability.

(2) Simplistic Predictors: The reliance on ECG and basic demographic data may overlook other potential predictors that could improve the models' accuracy and predictive power.

(3) Regional Specificity: The study's cohort is limited to Tibet, and the findings may not be directly applicable to other high-altitude populations without further validation.

Comments on revised version:

The authors have made revisions in response to the primary concerns raised in the initial review, leading to significant improvements in the manuscript's technical accuracy, formatting consistency, and overall clarity. They have provided a detailed explanation of the selection criteria for the final model variables, which has enhanced the transparency and robustness of the study's methodology. Additionally, the authors have acknowledged the limitation of lacking external validation in cohorts from other high-altitude regions and outlined their plans for future research to address this issue.

---

## [Author Response]

The following is the authors’ response to the original reviews.

**Reviewer #1:**
(1) Correct capitalization errors, ensuring the first letter of each sentence is capitalized.

Thank you for your comment. We have corrected capitalization errors.

(2) Ensure that all technical terms and abbreviations are introduced in full when first mentioned and consistently used throughout the text.

Thank you for your comment. we have checked and corrected the issue.

(3) Review the manuscript for grammatical errors and improve sentence structures to enhance readability.

Thank you for your comment. we have checked and corrected the issue.

(4) Ensure all figures referenced in the text, such as Fig. 3G, are appropriately discussed and integrated into the narrative.

Thank you for your comment. we have discussed and integrated Fig. 3G into the narrative (Page 12, Line 162-166).

(5) Maintain consistent formatting, including first-line indentation and spacing before paragraphs, to improve the document's visual coherence.

Thank you for your comment. we have checked and corrected the issue.

(6) Provide additional explanations for the selection criteria of final model variables, particularly the rationale behind choosing the λ_1se criterion in the LASSO regression.

Thank you for your comment. we have provided explanations for choosing the λ_1se criterion in the LASSO regression (Page 25, Line 315-316; Page 27, Line 363-364).

(7) Conduct validation studies with cohorts from other high-altitude regions to assess the generalizability and robustness of the prediction models.

Thank you for your comment. The lack of validation of cohorts from other high-altitude regions is a weakness in this study, and in our follow-up study, we will conduct external validation with cohorts from more other high-altitude regions to assess the generalizability and robustness of our prediction models.